# Slimming the Giant: Efficient Structured Pruning for Adapter-Tuned SAM

## Abstract

Foundation models with parameter-efficient adapters enable strong segmentation but remain hard to deploy due to scale and cost. We propose Adapter-aware Structured Sparsification (ASSP), a structured pruning framework for adapter-tuned SAM. ASSP begins with a concise dependency analysis of backbone–adapter couplings and derives unified slicing rules for heads, channels, and kernels. It then scores structures via a Projected-Gradient Residual criterion that aligns upstream and downstream gradient subspaces, and restores accuracy with a dual-stream compensation scheme that alternates supervision on both data sources. The procedure runs in two stages: prune and recover adapters, then freeze adapters and prune and recover the backbone. Built on SAM-Med2D, ASSP uses only 20k images (0.4% of SA-Med2D-20M) yet reduces encoder parameters by over 75% and compute to about one quarter, while Dice typically stays within two points of the baseline. Under the same calibration budget it outperforms a transferred SlimSAM baseline and yields consistent latency and throughput gains on H20 GPUs. Although evaluated on medical data, the dependency modeling, PGR scoring, and dual-stream compensation are task-agnostic and broadly applicable to adapter-tuned models.

## 1 Introduction

Since the release of Segment Anything (SAM) (Kirillov et al., 2023), its strong performance and generalization have made it a common foundation for fine-tuning across segmentation tasks. In natural images, PerSAM and PerSAM-F (Zhang et al., 2023b) personalize SAM without updating its weights, needing only a single image-mask pair and even improving DreamBooth outputs (Ruiz et al., 2023). In medical imaging, MedSAM (Ma et al., 2024a) fully fine-tunes (except the prompt encoder) and attains strong results across 10 modalities and 30+ cancer types, while Medical SAM Adapter (Wu et al., 2025) uses adapter-based parameter-efficient tuning to reach SOTA after dataset-specific finetuning. SAM-Med3D (Wang et al., 2024) and SAM-Med2D (Cheng et al., 2023) adapt SAM to 3D and 2D medical domains via adapters, enabling general-purpose medical segmentation; in remote sensing, SGO/SGB (Ma et al., 2024b) exploits SAM-derived object/boundary cues with new losses to boost semantic segmentation. As a result, adapter-based fine-tuning has become a prevalent recipe. Yet SAM-family models remain heavy, and adapters can exacerbate deployment cost, e.g., SAM-Med2D's adapter parameters exceed those of the base SAM, forcing reduced input resolution to sustain throughput (Cheng et al., 2023). This motivates compression methods tailored to adapter-tuned SAM.

Recent compression efforts target three axes: architecture redesign, quantization, and pruning. Fast-SAM (Zhao et al., 2023) replaces SAM's ViT with a CNN and reframes "segment anything" as instance segmentation trained on an SA-1B subset, while MobileSAM (Zhang et al., 2023a) distills a lighter ViT-based model. For quantization, general ViT methods exist (Li et al., 2023; Lin et al., 2021), and PQ-SAM (Liu et al., 2024) introduces learnable shifts/scales and outlier truncation specifically for SAM. For pruning, SlimSAM (Chen et al., 2024) combines a two-stage strategy with distillation to reach SOTA. However, despite the widespread use of adapters in downstream finetuning, there is still no compression approach explicitly designed for adapter-tuned SAM models. To explore efficient compression under adapter-based fine-tuning, we use the general-purpose medical segmentation model SAM-Med2D as our base model (Cheng et al., 2023).

Our first step towards an effective compression algorithm is to analyze the intrinsic properties of the hybrid backbone-adapter architecture. Specifically, we investigate the asymmetric sensitivity to parameter perturbations between the frozen backbone, which encodes general-purpose representations, and the trainable adapters, which specialize in the downstream task. Intuitively, the adapter processes the backbone's output; consequently, any perturbations in the backbone are propagated and amplified, whereas perturbations within the adapter are mitigated by the residual connection to the stable backbone activations. This leads us to hypothesize that adapters are inherently more robust to compression than the backbone.

To validate this hypothesis, we benchmark on SAM-Med2D and emulate the effects of pruning by applying random masking with sparsity increasing from 0% to 50% to the backbone and adapter weights independently. We evaluate the performance degradation on the corresponding SA-Med2D-20M dataset. As shown in Figure 1, the results measured by the Dice coefficient are stark: sparsifying the backbone incurs a significantly steeper performance degradation than sparsifying the adapter. This confirms that adapters are substantially more tolerant to parameter removal. This core observation forms the foundation of our subsequent method design.

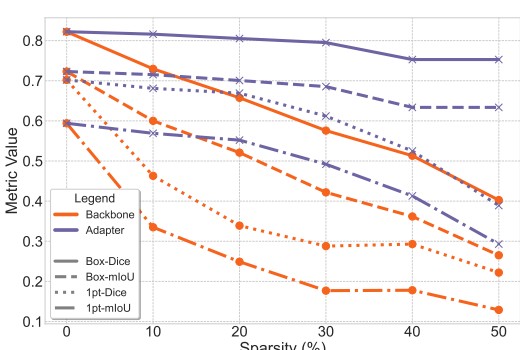

Figure 1: Backbone vs. Adapter Random-Mask Sparsity Sensitivity (0-50%)

Motivated by the above observations, this paper introduces ASSP, an iterative pruning method specifically tailored for SAM models fine-tuned with Adapters. Each iteration of ASSP consists of two distinct stages: pruning and compensation. In the pruning stage, we assess the consistency of gradient directions obtained from small-scale pretraining and downstream datasets for both backbone and adapter modules, identifying redundant parameter subsets exhibiting similar behavior. In the compensation stage, we employ an anchored fine-tuning strategy using both datasets, thereby mitigating the tendency of the model to overfit on small-scale data. Our primary contributions can be summarized as follows:

- **Systematic dependency analysis.** We analyze local dependencies and residual-induced couplings in SAM-family encoders and, based on these findings, propose ASSP, an adapter-aware structured pruning framework that applies consistent row and column slicing, aligns input and output channels, and uses equal per-head channel pruning.

- **PGR importance scoring.** We introduce a Projected Gradient Residual criterion that fuses upstream and downstream gradient subspaces with a Gram-based merge strategy and two-fold leave-one-out, then ranks heads, channels, and kernels by residual energy for structure-level pruning.

- **Dual-stream compensation.** We design a two-stage dual-stream compensation scheme. Stage 1 prunes and compensates adapters. Stage 2 prunes and compensates the backbone. Training alternates mini-batches from upstream and downstream data with balanced distillation. The approach delivers large reductions in parameters and MACs with near-lossless Dice using only 20K images, which is 0.4% of SA-Med2D-20M.

All implementation code is included in the supplementary materials.

## 2 RELATED WORK

### 2.1 PRUNING

Pruning is a crucial model compression technique typically classified into one-shot and multi-shot approaches based on pruning frequency. Originating from early methods such as Optimal Brain Damage (LeCun et al., 1989) and Optimal Brain Surgeon (Hassibi & Stork, 1992), one-shot pruning aims to reduce model sparsity to a targeted level in a single step. Recent one-shot pruning methods such as Wanda (Sun et al., 2023), SparseGPT Frantar & Alistarh (2023), and ICP (Luo et al.,

2025) have demonstrated considerable effectiveness on large language models (LLMs). However, vision models generally have smaller scales compared to LLMs; thus, multi-shot pruning methods, which iteratively prune and fine-tune models, typically yield superior performance for vision models like SAM. However, due to the vast scale of pretraining datasets used in foundational vision models, researchers often utilize a limited subset of pretraining data during the error compensation (fine-tuning) stage. For instance, SlimSAM proposes a multi-shot pruning approach that employs only 0.1% of the original pretraining dataset. SlimSAM divides the pruning process into two stages, leveraging intermediate-layer distillation techniques, and achieves better performance compared to FastSAM, MobileSAM, and EfficientSAM, even though the model sizes are comparable. Nevertheless, the aforementioned pruning methods lack specific optimization strategies tailored for adapter-based fine-tuning scenarios.

## 2.2 SAM-MED2D

SAM-Med2D fine-tunes SAM for 2D medical segmentation by fully updating the mask decoder and adapting the image encoder with inserted adapters, trained on a large composite of public datasets with over 4.6M images and about 19.7M masks. This yields a decoupled parameterization where the pretrained backbone remains shared and the adapters inject task-specific knowledge, and it produces a clear parameter and compute split as shown in

Table 1: SAM-Med2D parameter and compute breakdown

| Model Part | Params (M) | MACs (G) |
|------------|-----------|----------|
| Non-adapter | 86.4 | 32.1 |
| Adapter | 180.5 | 33.1 |
| Total | 267.0 | 65.0 |

Table 1: adapters contain 180.5M parameters and account for 33.1G MACs, the non-adapter part contains 86.4M parameters and accounts for 32.1G MACs, and the total model has 267.0M parameters and 65.0G MACs. We choose SAM-Med2D because this decoupling and asymmetry match our setting: PGR contrasts pretraining and downstream subspaces, and our pruning and compensation are adapter-aware while preserving the backbone's shared representation.

## 2.3 ADAPTER TUNING

Adapter tuning is one of the prevalent parameter-efficient fine-tuning methods, inserting additional trainable modules into pre-trained models to inject task-specific knowledge without disrupting the pretrained weights. Compared to LoRA (Hu et al., 2022; Dettmers et al., 2023), Adapter Tuning (Houlsby et al., 2019; Xia et al., 2022) strictly introduces new modules without altering the original backbone parameters, resulting in a high degree of decoupling between knowledge from pre-training and downstream tasks. Additionally, Adapter modules exhibit plug-and-play flexibility. These attributes constitute essential advantages that our pruning method effectively leverages.

## 3 METHOD

### 3.1 OVERVIEW

We address adapter-tuned SAM by proposing a deployment-oriented structured pruning strategy that maximally reduces model size and parameter count while preserving downstream accuracy under high-throughput or compute-constrained settings. To this end, we introduce ASSP, an adapter-aware structured sparsification framework. The design rests on two observations: (i) the frozen backbone is markedly more sensitive to pruning than the adapters; and (ii) adapters are integrated as residual bypasses, enabling stronger absorption of upstream perturbations. Building on these, we propose a gradient-projection residual importance metric that quantifies the directional consistency of gradients computed on a small pretraining subset and on downstream data; the residual after projection exposes redundant behaviors, which are pruned at appropriate structured granularities (e.g., channels or attention heads). We further pair this with a decoupled compensation mechanism—an anchored fine-tuning procedure that jointly leverages both data sources on backbone, to mitigate small-sample overfitting and recover accuracy. Implementation details follow.

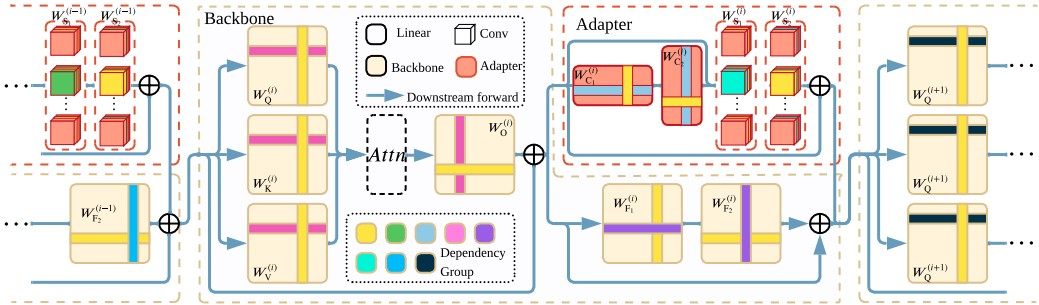

Figure 2: Dependency schematic. Shown for encoder layers $i-1$, $i$, and $i+1$ of the image encoder, the diagram illustrates how different components couple. Colored highlights denote individual rows or columns; entries sharing the same color constitute a dependency group that must be retained or pruned jointly.

## 3.2 DEPENDENCY ANALYSIS

We use *dependency* to mean the *structured co-keep/co-prune constraint* induced by operator-dimension matching inside two-layer bottlenecks and by residual-add shape alignment across branches. In Figure 2, colored highlights mark rows/columns that must move together as a *basic pruning unit*: magenta = MHA projection coupling ($Q/K/V \leftrightarrow O$), purple = FFN bottleneck ($F_1 \leftrightarrow F_2$), cyan = adapter-channel bottleneck ($C_1 \leftrightarrow C_2$), turquoise = adapter-spatial/conv bottleneck ($S_1 \leftrightarrow S_2$), and yellow = residual-alignment edges (global coupling). A more comprehensive and detailed dependency analysis is presented in Appx. A.1.

**Global vs. local granularity.** *Global dependencies* arise *only* when pruning the *second* operator's **output** (e.g., $O, F_2, C_2, S_2$; yellow): the residual add then forces a shared retained set $\mathcal{K} \subseteq \{1, \ldots, d_{\text{model}}\}$ that must propagate to all *first* operators in block $i+1$ ($Q/K/V$, $F_1$, $C_1$, $S_1$). This synchronization yields *coarser* pruning granularity and tends to amplify ranking noise, so we deliberately *minimize* global coupling. In contrast, *local dependencies* close within a branch (purple, cyan, turquoise): pruning the **output columns** of the first operator must be paired with pruning the **input rows/channels** of the second, enabling *finer* units and more stable scoring.

**Practical units for MHA/FFN/Adapters.** In MHA we adopt *equal per-head channel pruning* (finer than whole-head pruning): if each head removes $c$ channels, we form $\mathcal{J}_{\text{attn}}^{(i)}$ and keep the same *columns* in $\boldsymbol{W}_{\text{Q}}^{(i)}, \boldsymbol{W}_{\text{K}}^{(i)}, \boldsymbol{W}_{\text{V}}^{(i)}$, while selecting the matching *rows* in $\boldsymbol{W}_{\text{O}}^{(i)}$ (the magenta group). With fused $\boldsymbol{W}_{\text{QKV}}$, we slice by a shifted-union over three segments (Appx.). This choice offers finer control than whole-head pruning (which removes in multiples of $d_h$) but requires resizing/reinitializing relative positional encodings after pruning. FFN and adapter bottlenecks follow the same paired pattern, e.g., $\boldsymbol{W}_{\text{F}_1}^{(i)}[:, \mathcal{J}_{\text{F}}^{(i)}]$ with $\boldsymbol{W}_{\text{F}_2}^{(i)}[\mathcal{J}_{\text{F}}^{(i)}, :]$, and analogously for channel/conv branches with $\mathcal{J}_{\text{C}}^{(i)}$ and $\mathcal{J}_{\text{S}}^{(i)}$.

In summary, the colored groups define the *atomic pruning units*: we prefer *local* units for finer granularity and reserve *global* alignment via $\mathcal{K}$ only when unavoidable. PGR in Section 3.3 scores exactly these locally closed units (per-head channel groups; linear/conv bottleneck pairs), and our compensation (Figure 4) distills at $O$ and $F_2$—the residual-alignment anchors—to preserve the shared subspace while keeping global synchronization minimal.

## 3.3 TASK-PROJECTED GRADIENT RESIDUAL

After adapter-based fine-tuning, the backbone and adapter parameters in SAM are highly decoupled: the backbone remains aligned with the upstream pretraining objective, while the adapter captures task-specific corrections for the downstream objective and interacts with the pretrained representations. Consequently, pruning must account for the fact that the downstream task consumes features produced by the upstream-trained backbone. We therefore propose a Projected-Gradient Residual (PGR) criterion to score structured units for pruning.

As outlined in Algorithm 1, we rely on first-order information because each pruning step is followed by a brief compensation phase. Concretely, we sample a small subset from the pretraining dataset and another from the downstream dataset. Using the image encoder's embeddings, we apply stochastic masking (to emulate structured removals) and compute gradients to obtain two gradient matrices: the pretraining gradients and the downstream gradients. As illustrated in Figure 2, we insert a skip path that bypasses the adapter during the pretraining forward/backward passes (to avoid adapter interference), while the downstream passes include both backbone and adapter. Thus, the backbone receives both pretraining and downstream gradients, whereas the adapter receives only downstream gradients.

To characterize dominant gradient directions per module, we build low-rank subspaces from unit-wise gradients. For the $i$-th module, let $\boldsymbol{G}_{\mathrm{pre}}^{(i)} \in \mathbb{R}^{n_u \times d}$ and $\boldsymbol{G}_{\mathrm{down}}^{(i)} \in \mathbb{R}^{n_u \times d}$ denote row-stacked gradients of all candidate structured units under the upstream and downstream subsets, respectively. We perform thin SVDs:

$$\boldsymbol{G}_{\mathrm{pre}}^{(i)} = \boldsymbol{W} \, \boldsymbol{\Sigma} \, \boldsymbol{V}^{\top}, \quad \boldsymbol{U}_{\mathrm{pre}}^{(i)} = \boldsymbol{V}_{[:,1:r]} \tag{1}$$

$$\boldsymbol{G}_{\mathrm{down}}^{(i)} = \hat{\boldsymbol{W}} \, \hat{\boldsymbol{\Sigma}} \, \hat{\boldsymbol{V}}^{\top}, \quad \boldsymbol{U}_{\mathrm{down}}^{(i)} = \hat{\boldsymbol{V}}_{[:,1:r]} \tag{2}$$

yielding rank-$r$ orthonormal bases that span the dominant gradient subspaces of the two tasks. Because downstream solutions typically reuse upstream features, we explicitly merge the two subspaces into a joint basis $\boldsymbol{U}_{\cup}^{(i)}$ using a numerically stable Gram-based procedure. We first concatenate the bases

$$\boldsymbol{C}^{(i)} = \left[\, \boldsymbol{U}_{\mathrm{pre}}^{(i)} \; \boldsymbol{U}_{\mathrm{down}}^{(i)} \,\right] \in \mathbb{R}^{d \times 2r} \tag{3}$$

form the Gram matrix $\boldsymbol{G}_C^{(i)} = \boldsymbol{C}^{(i)^{\top}} \boldsymbol{C}^{(i)} = \boldsymbol{V}_C \, \boldsymbol{\Lambda}_C \, \boldsymbol{V}_C^{\top}$, select $k \leq 2r$ by a fixed rank or energy threshold, and obtain the merged orthonormal basis

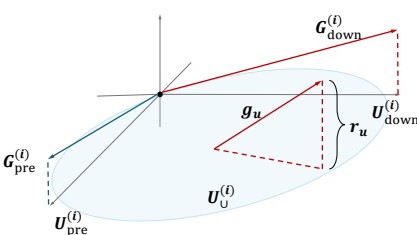

Figure 3: PGR geometry: project a unit gradient $\boldsymbol{g}_u$ onto the merged subspace $\boldsymbol{U}_{\cup}$; the normal (off-plane) residual $\boldsymbol{g}_u - \Pi_{\cup}(\boldsymbol{g}_u)$ gives the PGR score.

$$\boldsymbol{U}_{\cup}^{(i)} = \boldsymbol{C}^{(i)} \, \boldsymbol{V}_{C[:,1:k]} \, \boldsymbol{\Lambda}_{C,1:k}^{-1/2}, \quad \boldsymbol{U}_{\cup}^{(i)^{\top}} \boldsymbol{U}_{\cup}^{(i)} = \boldsymbol{I}_k \tag{4}$$

This $\boldsymbol{U}_{\cup}^{(i)}$ represents the jointly explainable directions spanned by upstream and downstream gradients within the module. In subsequent steps, we project the gradient of each unit on $\boldsymbol{U}_{\cup}^{(i)}$ and use the residual energy - interpreted as the part that is not reproducible by other units along the consistent directions of the task - to define the importance of PGR for structured pruning. Importantly, when building $\boldsymbol{G}_{\mathrm{pre}}^{(i)}$, we bypass the adapter (backbone only), whereas $\boldsymbol{G}_{\mathrm{down}}^{(i)}$ includes both the backbone and the adapter. This design faithfully captures the dependency structure of the two objectives and avoids leakage between them. Directly projecting $\boldsymbol{G}_{\mathrm{down}}^{(i)}$ onto an estimated basis is optimistically biased. We therefore adopt a two-fold scheme. Let the row indices of $\boldsymbol{G}_{\mathrm{down}}^{(i)}$ be partitioned into disjoint sets $\mathsf{S}_1, \mathsf{S}_2$. Build fold-specific merged bases $\boldsymbol{U}_{\cup,[1]}^{(i)}$ and $\boldsymbol{U}_{\cup,[2]}^{(i)}$ by merging $\boldsymbol{U}_{\mathrm{pre}}^{(i)}$ with the SVD basis computed from the rows in $\mathsf{S}_1$ and $\mathsf{S}_2$ respectively. The residual energy for unit $u$ is

$$R_u^2 \;=\; \|\boldsymbol{g}_u\|_2^2 - \|\boldsymbol{U}_{\cup,[-u]}^{(i)^{\top}} \boldsymbol{g}_u\|_2^2 \;\approx\; \begin{cases} \|\boldsymbol{g}_u\|_2^2 - \|\boldsymbol{U}_{\cup,[2]}^{(i)^{\top}} \boldsymbol{g}_u\|_2^2, & u \in \mathsf{S}_1, \\ \|\boldsymbol{g}_u\|_2^2 - \|\boldsymbol{U}_{\cup,[1]}^{(i)^{\top}} \boldsymbol{g}_u\|_2^2, & u \in \mathsf{S}_2 \,. \end{cases} \tag{5}$$

and we set $r_u = \sqrt{\max\left(R_u^2, 0\right)}$. Intuitively, $r_u$ measures the component of $\boldsymbol{g}_u$ that cannot be represented by task-consistent directions estimated without using $\boldsymbol{g}_u$ itself. Then, we convert residuals to pruning scores with a scale-aware normalization:

$$s_u = r_u \cdot \|\boldsymbol{\theta}_u\|_2 \tag{6}$$

where $\boldsymbol{\theta}_u$ is the parameter vector of unit $u$.

PGR differs from magnitude or Taylor criteria by explicitly separating substitutable versus non-substitutable gradient components under multi-task coupling (upstream + downstream). Units whose

gradients lie largely inside the merged subspace $\boldsymbol{U}_{\mathrm{pre}}^{(i)}$ are deemed redundant because their effect can be reproduced by remaining units during compensation; units with large residuals are preserved. This mechanism is agnostic to whether the module is backbone (ViT) or adapter (CNN/MLP) and naturally respects structured dependencies.

---

**Algorithm 1** Projected-Gradient Residual (PGR) *Scoring* for Structured Pruning

---

**Input**: Per-block unit gradients $\{\boldsymbol{G}_{\mathrm{pre}}^{(i)}, \boldsymbol{G}_{\mathrm{down}}^{(i)}\}_{i=1}^{n}$
**Parameter**: rank $r$, thresholds $\tau$; (optional) target sparsity $\rho$, safety constraints
**Output**: Per-unit scores $\{s_u\}$ for all candidate structured units

1: **for** $i = 1$ **to** $n$ **do**
2:   **(Bases by SVD)** $\boldsymbol{G}_{\mathrm{pre}}^{(i)} = \boldsymbol{W}\boldsymbol{\Sigma}\boldsymbol{V}^{\top}$, set $\boldsymbol{U}_{\mathrm{pre}}^{(i)} \leftarrow \boldsymbol{V}_{[:,1:r]}$;   $\boldsymbol{G}_{\mathrm{down}}^{(i)} = \hat{\boldsymbol{W}}\hat{\boldsymbol{\Sigma}}\hat{\boldsymbol{V}}^{\top}$, set $\boldsymbol{U}_{\mathrm{down}}^{(i)} \leftarrow \hat{\boldsymbol{V}}_{[:,1:r]}$.
3:   **(Merge via Gram)** $\boldsymbol{C}^{(i)} \leftarrow [\boldsymbol{U}_{\mathrm{pre}}^{(i)}\ \boldsymbol{U}_{\mathrm{down}}^{(i)}]$; $\boldsymbol{G}_{C}^{(i)} \leftarrow \boldsymbol{C}^{(i)\top}\boldsymbol{C}^{(i)} = \boldsymbol{V}_C\boldsymbol{\Lambda}_C\boldsymbol{V}_C^{\top}$; $k \leftarrow \min\big(2r, \#\{j : \lambda_{C,j} > \tau\}\big)$; $\boldsymbol{U}_{\cup}^{(i)} \leftarrow \boldsymbol{C}^{(i)}\boldsymbol{V}_{C[:,1:k]}\boldsymbol{\Lambda}_{C,1:k}^{-1/2}$.
4:   **(Two-fold LOO split)** Partition row indices of $\boldsymbol{G}_{\mathrm{down}}^{(i)}$ into disjoint $\mathsf{S}_1, \mathsf{S}_2$.
5:   **(Fold-1 basis)** $\boldsymbol{G}_{[1]} = \tilde{\boldsymbol{W}}_1\tilde{\boldsymbol{\Sigma}}_1\tilde{\boldsymbol{V}}_1^{\top}$, $\boldsymbol{U}_{[1]}^{\downarrow} \leftarrow \tilde{\boldsymbol{V}}_{1[:,1:r]}$; $\boldsymbol{C}_{[1]}^{(i)} \leftarrow [\boldsymbol{U}_{\mathrm{pre}}^{(i)}\ \boldsymbol{U}_{[1]}^{\downarrow}]$; $\boldsymbol{G}_{C_1}^{(i)} \leftarrow \boldsymbol{C}_{[1]}^{(i)\top}\boldsymbol{C}_{[1]}^{(i)} = \boldsymbol{V}_{C_1}\boldsymbol{\Lambda}_{C_1}\boldsymbol{V}_{C_1}^{\top}$; $k_1 \leftarrow \min(r, \#\{\lambda_{C_1} > \tau\})$; $\boldsymbol{U}_{\cup,[1]}^{(i)} \leftarrow \boldsymbol{C}_{[1]}^{(i)}\boldsymbol{V}_{C_1[:,1:k_1]}\boldsymbol{\Lambda}_{C_1,1:k_1}^{-1/2}$.
6:   **(Fold-2 basis)** $\boldsymbol{G}_{[2]} = \tilde{\boldsymbol{W}}_2\tilde{\boldsymbol{\Sigma}}_2\tilde{\boldsymbol{V}}_2^{\top}$, $\boldsymbol{U}_{[2]}^{\downarrow} \leftarrow \tilde{\boldsymbol{V}}_{2[:,1:r]}$; $\boldsymbol{C}_{[2]}^{(i)} \leftarrow [\boldsymbol{U}_{\mathrm{pre}}^{(i)}\ \boldsymbol{U}_{[2]}^{\downarrow}]$; $\boldsymbol{G}_{C_2}^{(i)} \leftarrow \boldsymbol{C}_{[2]}^{(i)\top}\boldsymbol{C}_{[2]}^{(i)} = \boldsymbol{V}_{C_2}\boldsymbol{\Lambda}_{C_2}\boldsymbol{V}_{C_2}^{\top}$; $k_2 \leftarrow \min(r, \#\{\lambda_{C_2} > \tau\})$; $\boldsymbol{U}_{\cup,[2]}^{(i)} \leftarrow \boldsymbol{C}_{[2]}^{(i)}\boldsymbol{V}_{C_2[:,1:k_2]}\boldsymbol{\Lambda}_{C_2,1:k_2}^{-1/2}$.
7:   **(Residuals)** $R_{[2]}^2 \leftarrow \|\boldsymbol{G}_{[2]}\|_{\mathrm{row}}^2 - \|(\boldsymbol{G}_{[2]}\boldsymbol{U}_{\cup,[1]}^{(i)})\|_{\mathrm{row}}^2$; $R_{[1]}^2 \leftarrow \|\boldsymbol{G}_{[1]}\|_{\mathrm{row}}^2 - \|(\boldsymbol{G}_{[1]}\boldsymbol{U}_{\cup,[2]}^{(i)})\|_{\mathrm{row}}^2$; assemble $R^2$ by placing $R_{[1]}^2$ on $\mathsf{S}_1$ and $R_{[2]}^2$ on $\mathsf{S}_2$, then set $r_u \leftarrow \sqrt{\max(R_u^2, 0)}$ for each unit $u$.
8:   **(Scoring)** $s_u \leftarrow r_u \cdot \|\boldsymbol{\theta}_u\|_2$
9: **end for**
10: **return** per-unit scores $\{s_u\}$

---

### 3.4 JOINT COMPENSATION WITH MULTI-STREAM DATA

PGR scores structured units by explicitly contrasting the *pretraining stream*, SA-1B, and the *downstream stream*, SA-Med2D-20M. To faithfully restore what PGR preserves after pruning, compensation must operate on the *same two streams*. We therefore design a two-stage, dual-stream scheme that mirrors the PGR-decoupled roles: the backbone is a shared feature extractor across the pretraining and downstream streams, while adapters provide task-specific refinements on top of it.

**Stage 1 — Adapter pruning and compensation.** We freeze the backbone and rank adapter units via PGR per dependency group defined by the Adapter Internal Dependencies. Groups are pruned to target sparsity $s$. This stage uses the downstream stream only—consistent with the PGR finding that adapters capture task-specific corrections on a frozen backbone—so early updates do not disturb the shared backbone subspace. For compensation, the pre-pruned model acts as the teacher and the pruned model as the student (Figure 4). For each adapter $i$, we impose (i) parameter-space consistency on $\boldsymbol{W}_{\mathrm{C}_2}$ and (ii) feature-space consistency on $\boldsymbol{W}_{\mathrm{S}_2}$, yielding losses $L_{\mathrm{C}_2}^{(i)}$ and $L_{\mathrm{S}_2}^{(i)}$, respectively. The stage objective is

$$L_{\mathrm{a}} = \lambda_{\mathrm{a}}\sum_{i=1}^{n}\big(L_{\mathrm{C}_2}^{(i)} + L_{\mathrm{S}_2}^{(i)}\big) + L_{\mathrm{task}}. \tag{7}$$

When computing $L_{\mathrm{task}}$, we account for the SAM-family's multi-point, iterative prompting: the task loss is averaged over multiple interaction steps within a query, mitigating degradation from single point/box supervision.

**Stage 2 — Backbone pruning and compensation.** Unlike Stage 1, we freeze all adapters and make the backbone trainable. Because adapters were fine-tuned with the backbone fully frozen, the

same backbone is shared by the pretraining and downstream streams at inference. To preserve the shared subspace that PGR relies on, we anchor supervision on both streams using small calibration sets $\mathcal{D}_{\text{pre}}$ and $\mathcal{D}_{\text{down}}$. For each iteration, rather than mixing samples from the two streams into a single mini-batch, we alternate mini-batches drawn separately from $\mathcal{D}_{\text{pre}}$ and $\mathcal{D}_{\text{down}}$; the supervision loss is the average over one pretraining mini-batch and one downstream mini-batch processed in tandem. The teacher is the pre-pruned model (backbone + frozen adapters), and the student is the backbone-pruned model with the same adapters frozen.

For each block $i$ we distill intermediate activations from $W_{\text{O}}^{(i)}$ and $W_{\text{F}_2}^{(i)}$ on both streams. Denote by $L_{\text{O}}^{(i)}$ and $L_{F_2}^{(i)}$ the per-block distillation losses. Given a pretraining mini-batch $\mathcal{B}_{\text{pre}}$ and a downstream mini-batch $\mathcal{B}_{\text{down}}$ in the same iteration, our Stage 2 objective is

$$L_{\text{b}} = \frac{\lambda_{\text{b}}}{2} \sum_{i=1}^{n} \left[ (L_{\text{O}}^{(i)} + L_{\text{F}_2}^{(i)})|_{\mathcal{B}_{\text{pre}}} + (L_{\text{O}}^{(i)} + L_{\text{F}_2}^{(i)})|_{\mathcal{B}_{\text{down}}} \right] + \frac{1}{2} \left( L_{\text{task}}^{(i)}|_{\mathcal{B}_{\text{pre}}} + L_{\text{task}}^{(i)}|_{\mathcal{B}_{\text{down}}} \right). \quad (8)$$

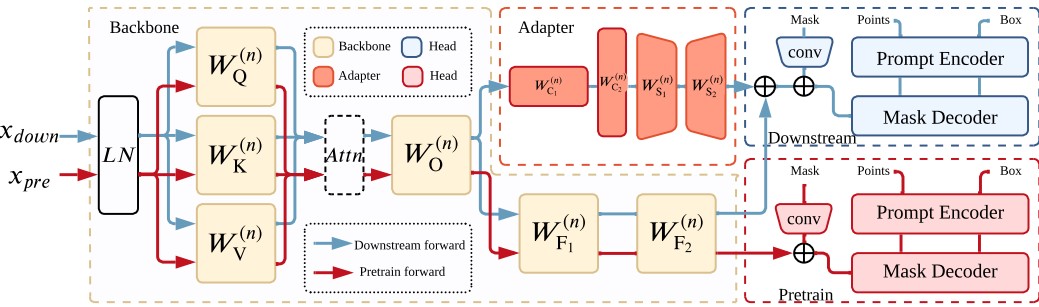

Figure 4: Overall compensation pipeline. Stage 1 prunes and compensates adapters with the backbone frozen (downstream-only); Stage 2 prunes and compensates the backbone with adapters frozen, using dual-stream supervision and distillation at $O$ and $F_2$ (residual-alignment anchors).

This alternating mini-batch schedule preserves data-parallel efficiency, avoids cross-stream mixing artifacts, and delivers balanced guidance from both sources. Together with dual-stream supervision, the two-stage design forms a tight loop with PGR: PGR separates substitutable from non-substitutable directions across the pretraining and downstream streams, and our compensation restores the shared subspace and residual-alignment anchors with stage-appropriate, data-efficient updates.

## 4 Experimental Results

### 4.1 Experimental Settings

We construct a 20k training set by uniformly sampling 10k images from SA-1B and 10k from SA-Med2D-20M, using a hash-based split that guarantees disjointness and reproducibility. Evaluation follows the official SAM-Med2D test protocol. The downstream task is interactive segmentation with point and box prompts. Each training sample uses seven interaction steps: the first step uses a box with probability 0.5 and the remaining steps use points. We report Dice for a single box and for 1, 3, and 5 points. Preprocessing and augmentation mirror the original SAM and SAM-Med2D pipelines. Upstream images are processed at $1024{\times}1024$ and downstream images at $256{\times}256$. The model is SAM-Med2D; pruning is applied only to the image encoder and all other components remain fixed. Pruning proceeds in two stages, adapters then backbone, with one-shot pruning per stage and local dependency groups; in multi-head attention we prune per-head channels rather than whole heads. PGR uses rank $r = 16$. The task loss combines focal, Dice, and mask IoU, and intermediate distillation uses mean squared error. We set $\lambda_{\text{a}} = 0.5$ and $\lambda_{\text{b}} = 0.5$. Optimization uses Adam with learning rate $1{\times}10^{-4}$, 40 epochs per stage, batch size 16 on four H800 GPUs, and `bf16` mixed precision. During backbone compensation, we alternate mini-batches drawn from the

upstream and downstream pools and average the two losses before each optimizer step with gradient accumulation enabled. MACs and parameter counts are reported for the encoder only at $256{\times}256$ input resolution. Implementation is in PyTorch.

## 4.2 Main Results

Structured pruning tailored to adapter-tuned models remains limited. We therefore compare ASSP primarily to the original SAM-Med2D and include a transferred baseline from SlimSAM adapted to our setting. For SlimSAM we use twenty thousand downstream images as the calibration set so that the total sample count matches our method, which uses ten thousand upstream images from SA-Med2D-20M and ten thousand downstream images from SAM-Med2D. In Table 2, the notation **A***xx* denotes the adapter pruned by *xx*% (i.e., *xx*% of adapter parameters removed) and **B***xx* denotes the backbone pruned by *xx*%; ASSP always prunes the adapter first and then the backbone. SM2D denotes SAM-Med2D. Finetune denotes the SAM model further fine-tuned on SA-Med2D-20M.

Table 2 shows that ASSP achieves large reductions in parameters and MACs with minimal accuracy loss. At moderate sparsity such as **A50** or **B50**, compute and parameter counts fall substantially while Dice under box and point prompts remains essentially unchanged. Under more aggressive compression such as **B75+A75**, MACs drop to roughly one quarter of the original and parameters decrease by more than seventy five percent, yet the single-box and multi-point Dice scores stay within about two points of the SAM-Med2D baseline. We also observe that pruning only the backbone, for example **B50**, yields smaller savings and a slightly larger accuracy drop than pruning only the adapter at a comparable target, for example **A50**, even though adapter pruning delivers greater reductions in parameters and MACs and thus a higher overall sparsity. Finally, ASSP attains these near-lossless trade-offs using only twenty thousand images in total, which is zero point four percent of the 4.6 million images in SA-Med2D-20M, underscoring the data efficiency of the approach.

Table 2: Main Result. We compare pruning applied to the backbone vs. the adapter. Notation: **B***xx*% means the **backbone** is pruned to sparsity *xx* (e.g., **B75** = backbone pruned to 0.75 sparsity); **A***xx* analogously denotes adapter pruning. SM2D denotes *SAM-Med2D*. Params and MACs are reported for the encoder

| Model | Strategy | Sparsity | Resolution | BBox (%) | 1pt (%) | 3pt (%) | 5pt (%) | Params (M) | MACs (G) |
|-------|----------|----------|------------|----------|---------|---------|---------|------------|----------|
| SAM | N/A | 0.00 | $256 \times 256$ | 61.63 | 18.94 | 28.28 | 37.47 | 89.67 | 32.07 |
| SAM | N/A | 0.00 | $1024 \times 1024$ | 74.49 | 36.88 | 42.76 | 47.57 | 89.67 | 368.86 |
| SM2D | N/A | 0.00 | $256 \times 256$ | 79.35 | 70.01 | 76.35 | 78.68 | 270.08 | 65.15 |
| SAM | Finetune | 0.00 | $256 \times 256$ | 73.56 | 60.11 | 70.95 | 75.51 | 89.67 | 32.07 |
| SM2D | SlimSAM | 0.50 | $256 \times 256$ | 74.10 | 65.23 | 74.64 | 76.93 | 134.43 | 32.75 |
| SM2D | B50 | 0.15 | $256 \times 256$ | **79.23** | **69.92** | **76.27** | **78.94** | **227.58** | **49.29** |
| SM2D | A50 | 0.33 | $256 \times 256$ | **79.32** | **69.43** | **76.26** | **78.84** | **179.88** | **48.63** |
| SM2D | B75 | 0.24 | $256 \times 256$ | **79.11** | **69.67** | **75.06** | **78.82** | **202.30** | **41.07** |
| SM2D | A75 | 0.50 | $256 \times 256$ | **79.39** | **68.81** | **75.78** | **78.98** | **134.76** | **40.36** |
| SM2D | A87.5 | 0.58 | $256 \times 256$ | **79.07** | **66.39** | **75.05** | **77.65** | **112.20** | **36.22** |
| SM2D | B50+A50 | 0.50 | $256 \times 256$ | **79.01** | **69.43** | **76.26** | **77.62** | **134.43** | **32.75** |
| SM2D | B75+A75 | 0.76 | $256 \times 256$ | **78.65** | **68.13** | **74.26** | **76.85** | **63.43** | **15.62** |

## 4.3 Ablation Study

**Ablation on Pruning Order.** Table 3 compares two pruning orders: adapter→backbone, denoted A→B, and backbone→adapter, denoted B→A. Across all sparsity targets and prompt settings, A→B consistently outperforms B→A. At B50+A50, A→B improves over B→A by 0.93 on BBox, 1.71 on 1pt, 0.71 on 3pt, and 1.59 on 5pt Dice. At B75+A75, the gains increase to 1.15 on BBox, 2.79 on 1pt, 1.80 on 3pt, and 0.91 on 5pt. Our interpretation is that A→B first compresses and compensates within local adapter dependencies, thereby preserving downstream domain capacity on the residual pathway. After freezing the adapter, backbone pruning with dual-stream compensation can realign the shared representation without compounding errors. In contrast, B→A perturbs the shared backbone subspace first and subsequently weakens alignment when the adapter is pruned,

leading to accumulated degradation. Based on these findings, we adopt A→B as the default pruning order in the main experiments, and retain this choice in subsequent studies.

Table 3: Ablation on pruning order. Dice (%) under box and 1/3/5-point prompts.

| Model | Order | BBox (%) | 1pt (%) | 3pt (%) | 5pt (%) |
|---|---|---|---|---|---|
| SM2D(B50+A50) | A→B | 79.01 | 69.43 | 76.26 | 77.62 |
| SM2D(B50+A50) | B→A | 78.08 | 67.72 | 75.55 | 76.03 |
| SM2D(B75+A75) | A→B | 78.65 | 68.13 | 74.26 | 76.85 |
| SM2D(B75+A75) | B→A | 77.50 | 65.34 | 72.46 | 75.94 |

**Ablation on Tuning Data Source.**   We study how the choice of calibration data affects recovery after pruning. Table 4 reports Dice under box and point prompts with the backbone pruned to 75% and a fixed budget of 20K images. Using only upstream data yields weaker alignment to downstream prompts, for example box 64.95, one point 52.47, three points 64.91, five points 71.74. Using only downstream data improves point prompts yet remains below the mixed setting on all metrics, for example box 68.31 and one point 64.69. Mixing sources with 10K upstream and 10K downstream produces the best accuracy across the board: box 79.23, one point 69.67, three points 75.06, five points 78.82. Gains are sizable relative to downstream only, for example +10.92 on box and +4.98 on one point, and even larger relative to upstream only. These results support the design of dual-stream supervision: upstream samples preserve the shared representation in the backbone while downstream samples supply task-specific signals, and their combination delivers the most reliable post-pruning recovery. We therefore adopt mixed tuning data in the main experiments.

Table 4: Effect of tuning data source at B75. Dice (%) under box and 1/3/5-point prompts.

| Tuning data | #Samples | BBox (%) | 1pt (%) | 3pt (%) | 5pt (%) |
|---|---|---|---|---|---|
| Upstream only | 20K | 64.95 | 52.47 | 64.91 | 71.74 |
| Downstream only | 20K | 68.31 | 64.69 | 73.45 | 78.02 |
| Mixed (Up+Down) | 10K+10K | **79.23** | **69.67** | **75.06** | **78.82** |

**Additional ablations.**   We include extended studies in Appx. A.2, covering sensitivity to the PGR rank and merge threshold, calibration budget and data composition, pruning granularity in MHA (per-head channels versus whole heads), robustness of the A→B order under different sparsity targets, variants of the alternating mini-batch schedule, and input-resolution effects. We also report encoder-only parameter and MAC accounting checks and qualitative error analyses to complement the quantitative results here.

## 5  CONCLUSION

We introduced ASSP, a structured pruning framework for adapter-tuned SAM family models, grounded in a systematic analysis of local and residual-induced dependencies. ASSP prunes with consistent row-column and $C_{in}/C_{out}$ slicing and uses equal per-head channel pruning. Central to our approach is the Projected-Gradient Residual (PGR) criterion, which explicitly merges upstream and downstream gradient subspaces and scores heads, channels, and kernels by residual energy. We pair PGR-guided pruning with a two-stage, dual-stream compensation scheme that first adapts the adapters and then the backbone; the adapter→backbone order and mixed upstream-downstream supervision yield robust recovery.

On SAM-Med2D, ASSP achieves large cuts in parameters and MACs with minimal Dice degradation using only 20K images, underscoring strong data efficiency. These results indicate that dependency-aware pruning coupled with PGR is an effective recipe for compressing adapter-tuned vision transformers. Future work will extend PGR and ASSP to broader adapter designs and backbones, explore combination with quantization and kernel-level optimizations, and evaluate across additional medical and non-medical benchmarks.

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

# A APPENDIX

## A.1 DEPENDENCY ANALYSIS

Structured pruning requires a careful analysis of dependencies between different model components (Fang et al., 2023). Naively removing channels without considering these couplings can lead to dimensional mismatches and render the model inoperable. In our discussion, we index the Transformer blocks by $i \in \{1, \ldots, n\}$. Each block operates on an embedding dimension of $d_{\text{model}}$, which in the attention mechanism is structured into $h$ heads, each of width $d_h$, such that $d_{\text{model}} = h \, d_h$. As illustrated in Figure 2, we categorize the dependencies into three types: **Backbone Internal Dependencies**, **Adapter Internal Dependencies**, and **Global Dependencies**.

### A.1.1 BACKBONE INTERNAL DEPENDENCIES

Backbone internal dependencies refer to the structural constraints within a single Transformer block ($i$), specifically within its Multi-Head Attention (MHA) and Feed-Forward Network (FFN) modules.

**MHA dependency.** We analyze dependencies rooted at pruning the output channels of $\boldsymbol{W}_{\text{Q}}^{(i)}$ (the same reasoning applies symmetrically to $\boldsymbol{W}_{\text{K}}^{(i)}$ and $\boldsymbol{W}_{\text{V}}^{(i)}$). As highlighted by the magenta dependency group in Figure 2, the components of the MHA module are tightly coupled. To maintain dimensional consistency for the attention score calculation ($\boldsymbol{q} \, \boldsymbol{k}^{\top}$) and subsequent operations, pruning the output channels of the query projection matrix ($\boldsymbol{W}_{\text{Q}}^{(i)}$) necessitates corresponding pruning on the key ($\boldsymbol{W}_{\text{K}}^{(i)}$) and value ($\boldsymbol{W}_{\text{V}}^{(i)}$) projection matrices. Consequently, the input channels of the output projection matrix ($\boldsymbol{W}_{\text{O}}^{(i)}$) must also be pruned to match the dimension of the concatenated outputs from the value heads. For layer $i$, denote by $\mathcal{J}_{\text{attn}}^{(i)} \subseteq \{1, \ldots, d_{\text{model}}\}$ the set of *retained*

Q/K/V output channels. We prune Q/K/V by keeping *columns* $\mathcal{J}_{\text{attn}}^{(i)}$ and prune the output projection $O$ by keeping the matching *rows* $\mathcal{J}_{\text{attn}}^{(i)}$:

$$\boldsymbol{W}_{\text{Q}}^{(i)}[:,\mathcal{J}_{\text{attn}}^{(i)}], \quad \boldsymbol{W}_{\text{K}}^{(i)}[:,\mathcal{J}_{\text{attn}}^{(i)}], \quad \boldsymbol{W}_{\text{V}}^{(i)}[:,\mathcal{J}_{\text{attn}}^{(i)}], \qquad \boldsymbol{W}_{\text{O}}^{(i)}[\mathcal{J}_{\text{attn}}^{(i)},:].$$

This coupling preserves both the inner-product dimension in $\boldsymbol{Q}\boldsymbol{K}^{\top}$ and the input dimension of $O$ after pruning. The two practical variants differ only in the construction of $\mathcal{J}_{\text{attn}}^{(i)}$ (here $[k] := \{1,\ldots,k\}$):

$$\mathcal{J}_{\text{attn}}^{(i)} = \begin{cases} \{(t-1)d_h + c \mid t \in \mathcal{H}^{(i)}, \, c \in [d_h]\}, & \textbf{Head pruning (whole heads)}, \\ \{(t-1)d_h + c \mid t \in [h], \, c \in \mathcal{C}^{(i)}\}, & \textbf{Equal per-head channel pruning}. \end{cases}$$

with $\mathcal{H}^{(i)} \subseteq [h]$ and $\mathcal{C}^{(i)} \subseteq [d_h]$. When using a fused projection $\boldsymbol{W}_{\text{QKV}}^{(i)} \in \mathbb{R}^{d_{\text{model}} \times 3d_{\text{model}}}$, let $d = d_{\text{model}}$ and define the shifted union $\mathcal{J}_{\text{attn}}^{(i,3)} = \mathcal{J}_{\text{attn}}^{(i)} \cup (\mathcal{J}_{\text{attn}}^{(i)} + d) \cup (\mathcal{J}_{\text{attn}}^{(i)} + 2d)$; pruning becomes

$$\boldsymbol{W}_{\text{QKV}}^{(i)}[:,\mathcal{J}_{\text{attn}}^{(i,3)}], \qquad \boldsymbol{W}_{\text{O}}^{(i)}[\mathcal{J}_{\text{attn}}^{(i)},:].$$

Although equal per-head channel pruning requires adjusting the shape of the relative positional encodings for compatibility, whereas head pruning does not, equal per-head channel pruning removes channels in multiples of $h$ (one per head), yielding a finer pruning granularity than head pruning, which removes channels in multiples of $d_h$ (entire heads). Therefore, in this work we prune $\boldsymbol{W}_{\text{Q}}, \boldsymbol{W}_{\text{K}}, \boldsymbol{W}_{\text{V}}$ using equal per-head channel pruning and re-initialize the relative positional encodings after pruning.

**Feed-forward network dependency.** As highlighted by the purple dependency group in Figure 2, pruning the output channels of $\boldsymbol{W}_{\text{F}_1}^{(i)}$ within the FFN is independent of the rest of the model and only couples to the internal $\boldsymbol{W}_{\text{F}_2}^{(i)}$. Let $\boldsymbol{W}_{\text{F}_1}^{(i)} \in \mathbb{R}^{d_{\text{model}} \times d_{\text{F}}}$ and $\boldsymbol{W}_{\text{F}_2}^{(i)} \in \mathbb{R}^{d_{\text{F}} \times d_{\text{model}}}$. With the retained hidden set $\mathcal{J}_{\text{F}}^{(i)} \subseteq [d_{\text{F}}]$, we prune as

$$\boldsymbol{W}_{\text{F}_1}^{(i)}[:,\mathcal{J}_{\text{F}}^{(i)}], \qquad \boldsymbol{W}_{\text{F}_2}^{(i)}[\mathcal{J}_{\text{F}}^{(i)},:],$$

### A.1.2 ADAPTER INTERNAL DEPENDENCIES

Unlike the backbone, adapters are often highly customized. In practice, one may mix linear and convolutional layers within the adapter to strengthen spatial feature modeling, at the cost of more intricate dependency patterns (Zhong et al., 2024; Chen et al., 2022; Wu et al., 2025; Cheng et al., 2023; Wang et al., 2024). Nevertheless, as with the FFN, adapter bottlenecks typically follow a paired design (e.g., *in-* and *out*-projections for linear layers, or down/up projections for convolutions), which induces *local* dependencies within the adapter rather than *global* couplings to the rest of the network. This is precisely the case in SAM-MED2D.

**Adapter-channel dependency.** We analyze dependencies rooted at pruning the output channels of $\boldsymbol{W}_{\text{C}_1}^{(i)}$. As highlighted by the cyan dependency group in Figure 2, and as with the FFN, the adapter's channel-focused bottleneck exhibits only a *local* dependency between its two linear layers: pruning the output channels of $\boldsymbol{W}_{\text{C}_1}^{(i)}$ merely requires pruning the matching input channels of $\boldsymbol{W}_{\text{C}_2}^{(i)}$, without inducing couplings to the rest of the model. Consequently, structured pruning typically operates by selecting the corresponding channel indices, with the finest granularity being a single channel-pair dependency. Denote the two channel-MLP matrices by $\boldsymbol{W}_{\text{C}_1}^{(i)} \in \mathbb{R}^{d_{\text{model}} \times d_{\text{ch}}}$ and $\boldsymbol{W}_{\text{C}_2}^{(i)} \in \mathbb{R}^{d_{\text{ch}} \times d_{\text{model}}}$, which produce a per-channel gate $x_w^{(i)} \in \mathbb{R}^{d_{\text{model}}}$. With $\mathcal{J}_{\text{C}}^{(i)} \subseteq [d_{\text{ch}}]$,

$$\boldsymbol{W}_{\text{C}_1}^{(i)}[:,\mathcal{J}_{\text{C}}^{(i)}], \qquad \boldsymbol{W}_{\text{C}_2}^{(i)}[\mathcal{J}_{\text{C}}^{(i)},:],$$

**Adapter-spatial dependency.** As highlighted by the turquoise dependency group in Figure 2, pruning in this branch involves *two* convolutions, with each kernel corresponding to a single output channel. Mirroring the FFN and adapter-channel cases, dependencies rooted at pruning the output channels of $\boldsymbol{W}_{\text{S}_1}^{(i)}$ are *local* to the adapter-spatial path and do not couple to the rest of the model.

Let the stride-2 convolution and the transposed convolution be $\boldsymbol{W}_{\mathrm{S}_1}^{(i)} \in \mathbb{R}^{C_{\mathrm{in}} \times C_{\mathrm{out}} \times 3 \times 3}$ and $\boldsymbol{W}_{\mathrm{S}_2}^{(i)} \in \mathbb{R}^{C_{\mathrm{in}} \times C_{\mathrm{out}} \times 4 \times 4}$ (*notation uses* $(C_{\mathrm{in}}, C_{\mathrm{out}}, k_h, k_w)$). Using the retained channel set $\mathcal{J}_{\mathrm{S}}^{(i)}$, we slice $\boldsymbol{W}_{\mathrm{S}_1}^{(i)}$ on its *output* channel axis and $\boldsymbol{W}_{\mathrm{S}_2}^{(i)}$ on its *input* channel axis:

$$\boldsymbol{W}_{\mathrm{S}_1}^{(i)}[:,\mathcal{J}_{\mathrm{S}}^{(i)},:,:], \qquad \boldsymbol{W}_{\mathrm{S}_2}^{(i)}[\mathcal{J}_{\mathrm{S}}^{(i)},:,:,:].$$

### A.1.3 GLOBAL DEPENDENCIES

Global dependencies arise primarily from the use of residual connections. In contrast to the *local* dependencies analyzed above—where pruning the *output* dimension of the first operator in each two-layer pair (e.g., $\boldsymbol{W}_{\mathrm{F}_1}$, $\boldsymbol{W}_{\mathrm{C}_1}$, $\boldsymbol{W}_{\mathrm{S}_1}$) does not trigger residual alignment—the situation changes when pruning the *output channels of the second operator*. As highlighted by the yellow group in Figure 2, both in the backbone and in the adapters, the abundance of residual additions forces cross-layer shape alignment whenever we prune the second operator in a pair. This mechanism effectively completes the remaining dependency edges of the image encoder. Starting from the output channels of the attention output projection as the root, let $\mathcal{K} \subseteq \{1, \ldots, d_{\mathrm{model}}\}$ denote the global set of retained channels at layer $i$. Because the residual add at block $i$ sums the outputs of attention, FFN, and adapter branches, we prune the *outputs* (columns / $C_{\mathrm{out}}$) of all second operators to the same set $\mathcal{K}$:

$$\boldsymbol{W}_{\mathrm{O}}^{(i)}[:,\mathcal{K}], \qquad \boldsymbol{W}_{\mathrm{F}_2}^{(i)}[:,\mathcal{K}], \qquad \boldsymbol{W}_{\mathrm{C}_2}^{(i)}[:,\mathcal{K}], \qquad \boldsymbol{W}_{\mathrm{S}_2}^{(i)}[:,\mathcal{K},:,:].$$

The block-$i$ residual output then exposes exactly the channel set $\mathcal{K}$ to block $i{+}1$. Consequently, we prune the *inputs* (rows / $C_{\mathrm{in}}$) of all first operators in block $i{+}1$ to $\mathcal{K}$:

$$\boldsymbol{W}_{\mathrm{Q}}^{(i+1)}[\mathcal{K},:], \quad \boldsymbol{W}_{\mathrm{K}}^{(i+1)}[\mathcal{K},:], \quad \boldsymbol{W}_{\mathrm{V}}^{(i+1)}[\mathcal{K},:],$$

$$\boldsymbol{W}_{\mathrm{F}_1}^{(i+1)}[\mathcal{K},:], \quad \boldsymbol{W}_{\mathrm{C}_1}^{(i+1)}[\mathcal{K},:], \quad \boldsymbol{W}_{\mathrm{S}_1}^{(i+1)}[\mathcal{K},:,:,:].$$

If a fused projection is used, the same input-row slicing applies to $\boldsymbol{W}_{\mathrm{QKV}}^{(i+1)}$, then use $\boldsymbol{W}_{\mathrm{QKV}}^{(i+1)}[\mathcal{K},:]$.

With a synchronized mask, the same $\mathcal{K}$ is reused across depth, yielding a single consistent channel subspace throughout the encoder. Moreover, residual connections propagate dependencies globally; under such global dependencies, the retained index set is also shared as a single $\mathcal{K}$ across modules and layers. This makes the pruning granularity substantially coarser than that of the local cases discussed earlier. Coarser granularity tends to amplify importance-ranking errors, and therefore our pruning procedure deliberately minimizes the use of global-dependency pruning whenever possible.

## A.2 ADDITIONAL ABLATION STUDY AND ANALYSIS

### A.2.1 ABLATION ON IMPORTANCE SCORE.

We compare convergence behavior and final accuracy across four structured importance criteria—Magnitude, Hessian, Taylor, and PGR. The convergence study tracks 1 point Dice over 40 epochs after pruning and compensation; Figure 5 plots the learning curves, and Table 5 reports the end of training Dice under box and point prompts. PGR consistently climbs faster and higher: its curve rises sharply within the first ten epochs and then stabilizes with low variance; Taylor follows at a small but steady gap; Magnitude improves more slowly and plateaus lower; Hessian trails throughout with the flattest slope. At convergence, PGR attains the best Dice on all prompts, with clear margins over the strongest non -PGR baseline in both single box and multi-point settings.

We attribute these gains to subspace alignment. PGR scores units by residual energy after projecting downstream gradients onto a joint upstream–downstream basis, suppressing directions that surviving units can reconstruct and preserving directions that are discriminative for both streams. This reduces gradient interference during recovery, yields cleaner updates, and shortens the compensation horizon. Empirically, we observe smoother learning with fewer oscillations, especially in early epochs when most recovery occurs. Taylor partially captures sensitivity yet lacks the joint subspace constraint, which explains its solid but second best curves. Magnitude ignores task geometry and thus requires longer fine tuning to approach a lower ceiling, while Hessian approximations are noisier under our data budget and yield the weakest recovery.

Overall, coupling pruning with task consistent gradient subspaces improves both final accuracy and convergence speed, enabling shorter compensation schedules at a fixed budget and making high sparsity settings more practical.

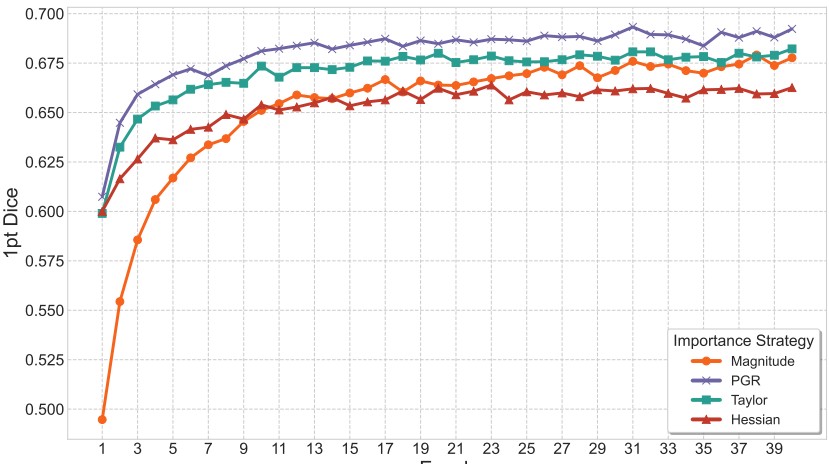

Figure 5: Convergence of 1-point Dice over 40 training epochs after pruning under four importance scores—Magnitude, Hessian, Taylor, and PGR. PGR rises fastest and stabilizes highest, Taylor follows, Magnitude improves more slowly, and Hessian lags. Curves are obtained with the alternating upstream–downstream compensation schedule.

Table 5: Comparison of importance scoring criteria. Metrics are Dice (%).

| Method | BBox (%) | 1pt (%) | 3pt (%) | 5pt (%) |
|---|---|---|---|---|
| Magnitude | 77.27 | 67.77 | 73.67 | 75.09 |
| Hessian | 77.92 | 66.26 | 73.84 | 77.08 |
| Taylor | 78.63 | 68.22 | 74.26 | 77.34 |
| PGR | **79.11** | **69.67** | **75.06** | **77.82** |

### A.2.2 LATENCY AND THROUGHPUT.

We evaluate at $256\times256$ with `bf16` in two modes: encoder only and end to end with encoder, prompt encoder, and mask decoder. Adapters are enabled, the main output fast path and QKV fusion are active, a single GPU is used, and no postprocessing is applied. Inference is measured on an NVIDIA H20. Training hardware is reported in Section 4.1. We sweep batch sizes $1, 4, 8$ and sparsities $0.00, 0.50, 0.75$, reporting mean, p50, and p90 latency in milliseconds and throughput in images per second, as shown in Tables 6 and 7.

On the encoder only path, pruning yields consistent acceleration. At batch size 1, mean latency improves from 22.493 ms to 19.779 ms, about $12\%$ faster, and throughput rises from 44.46 to 50.56 images per second, about $14\%$ higher. At batch size 8, mean latency drops from 35.025 ms to 27.956 ms, about $20\%$ faster, and throughput increases from 228.41 to 286.16 images per second, about $25\%$ higher. The gap between p50 and p90 remains small, indicating stable runtimes.

End to end results show similar gains, moderated by the unpruned prompt and mask heads. At batch size 1, mean latency decreases from 28.174 ms to 25.343 ms, about $10\%$ faster, and throughput improves from 35.49 to 39.46 images per second, about $11\%$ higher. At batch size 8, mean latency reduces from 41.179 ms to 33.961 ms, about $18\%$ faster, and throughput rises from 194.27 to 235.56 images per second, about $21\%$ higher. Variability remains low across percentiles.

These measurements confirm that encoder structured sparsity converts directly into practical speed and throughput improvements even on a baseline with QKV fusion. Benefits grow with batch size

due to better operator utilization and amortized launch overhead, and end to end gains remain double digit despite fixed costs outside the encoder.

Table 6: Encoder-only latency and throughput at $256 \times 256$ with `bf16` on NVIDIA H20. We report mean, p50, and p90 latency (ms) and throughput (img/s) across batch sizes $1, 4, 8$ and sparsity levels $0.00, 0.50, 0.75$.

| BS | Sparsity | Mean (ms) | p50 (ms) | p90 (ms) | Throughput (img/s) |
|---|---|---|---|---|---|
|   | 0.00 | 22.493 | 22.472 | 22.524 | 44.46 |
| 1 | 0.50 | 20.979 | 20.949 | 21.035 | 47.67 |
|   | 0.75 | 19.779 | 19.748 | 19.816 | 50.56 |
|   | 0.00 | 27.861 | 27.815 | 27.906 | 143.57 |
| 4 | 0.50 | 24.820 | 24.801 | 24.872 | 161.16 |
|   | 0.75 | 23.543 | 23.522 | 23.582 | 169.90 |
|   | 0.00 | 35.025 | 34.993 | 35.073 | 228.41 |
| 8 | 0.50 | 29.865 | 29.841 | 29.903 | 267.87 |
|   | 0.75 | 27.956 | 27.925 | 27.994 | 286.16 |

Table 7: End-to-end latency and throughput at $256 \times 256$ with `bf16` on NVIDIA H20 under single-point prompting. We report mean, p50, and p90 latency (ms) and throughput (img/s) across batch sizes $1, 4, 8$ and sparsity levels $0.00, 0.50, 0.75$.

| BS | Sparsity | Mean (ms) | p50 (ms) | p90 (ms) | Throughput (img/s) |
|---|---|---|---|---|---|
|   | 0.00 | 28.174 | 28.116 | 28.202 | 35.49 |
| 1 | 0.50 | 26.629 | 26.609 | 26.671 | 37.55 |
|   | 0.75 | 25.343 | 25.258 | 25.324 | 39.46 |
|   | 0.00 | 33.815 | 33.804 | 33.898 | 118.29 |
| 4 | 0.50 | 30.747 | 30.717 | 30.800 | 130.09 |
|   | 0.75 | 29.441 | 29.404 | 29.489 | 135.86 |
|   | 0.00 | 41.179 | 41.119 | 41.275 | 194.27 |
| 8 | 0.50 | 36.001 | 35.984 | 36.091 | 222.22 |
|   | 0.75 | 33.961 | 33.933 | 34.081 | 235.56 |

ETHICS STATEMENT

This work adheres to the ICLR Code of Ethics. No human subjects or animal experiments were involved. All datasets are publicly available and were used in compliance with their licenses and usage guidelines: the upstream data come from **SA-1B**, and the downstream data from **SA-Med2D-20M** (evaluation follows its official interactive segmentation protocol). No personally identifiable information was used, and no experiments were conducted that could raise privacy or security concerns. We have aimed to avoid biased or discriminatory outcomes and to maintain transparency and integrity throughout the research process.

REPRODUCIBILITY STATEMENT

We provide an anonymous repository with all code and configurations to enable verification and replication. The paper and appendix describe the full setup, including: a two-stage structured pruning pipeline (*adapter first, then backbone*); pruning applied only to the *image encoder* with reporting of MACs/parameters on the encoder at $256 \times 256$; local dependency groups with equal per-head channel pruning in attention; the PGR implementation (rank $r=16$, Gram-based merge with two-fold leave-one-out); dual-stream compensation with alternating mini-batches from SA-1B and SA-Med2D-20M and averaged losses per optimizer step; loss composition (focal + Dice + mask-IoU for the task loss, MSE for intermediate distillation) with weights $\lambda_a=0.5$ and $\lambda_b=0.5$; input resolutions

$1024\times1024$ for upstream and $256\times256$ for downstream. Optimization uses Adam with learning rate $1\times10^{-4}$, 40 epochs per stage, batch size 16, and `bf16` mixed precision. Implementation is in PyTorch. Training was run on NVIDIA H800 GPUs; inference profiling was run on NVIDIA H20 GPUs. Additional ablations and dependency details are provided in the appendix to facilitate faithful reproduction and extension.

## LLM USAGE

A large language model (LLM) was used solely for language editing, including wording refinement, grammar checking, and improving readability. The LLM did not participate in idea conception, methodological design, experiment implementation, or data analysis. The authors take full responsibility for the scientific content, and ensured that any LLM-assisted text complies with academic ethics and does not constitute plagiarism or scientific misconduct.

