# OpenReview forum: "Slimming the Giant: Efficient Structured Pruning for Adapter-Tuned SAM"
_ICLR.cc/2026/Conference — ICLR 2026 Conference Withdrawn Submission_

### Official Review · Reviewer_UGjS · 2025-10-28

**Soundness:** 2
**Presentation:** 1
**Contribution:** 1
**Rating:** 2
**Confidence:** 2

**Summary:**

The paper presents a method to make large adapter-tuned SAM models more efficient by pruning unnecessary parts of the network. The proposed approach, ASSP, removes redundant parameters and at the same time try to keep accuracy (almost) unchanged. It works in two stages: First, simplify the adapters, then the backbone. Experiments on medical image segmentation show that the method cuts model size and computation by about 75% with only minor drops in performance.

**Strengths:**

The paper addresses an important topic, namely improving the efficiency of large adapter-based segmentation models. The overall motivation is reasonable. The idea of pruning adapters and backbone in separate stages is intuitively sound, and the experiments demonstrate that some reduction in model size is possible without severe performance loss.

**Weaknesses:**

The paper is poorly presented and difficult to interpret. Although it introduces many terms and claims to perform “structured pruning,” the underlying ideas are only loosely explained and seem largely heuristic. Key concepts, such as “dependency analysis” and “projected gradient residual,” are described in vague, abstract language without sufficient intuition or mathematical grounding. For example, statements like “We use dependency to mean the structured co-keep/co-prune constraint induced by operator dimension matching inside two-layer bottlenecks and by residual-add shape alignment across branches” are not clearly defined and make it hard to understand what is actually being done. Overall, the method appears ad hoc, with unclear motivation for its design choices. The experiments show modest gains, but given the lack of clarity and justification, it is difficult to assess the soundness or reproducibility of the approach.

Another weakness is that the framework is only applied to a particular model for medical images. Why not other models?

**Questions:**

Could the authors provide a clearer and more intuitive explanation of what is actually meant by “dependency analysis”? In particular, what does the “co-keep/co-prune constraint induced by operator dimension matching” practically imply for the pruning process?

How is the proposed “Projected Gradient Residual (PGR)” criterion fundamentally different from existing gradient- or magnitude-based pruning heuristics? A mathematical or empirical comparison would be helpful.

---

### Official Review · Reviewer_4pHx · 2025-11-01

**Soundness:** 3
**Presentation:** 2
**Contribution:** 3
**Rating:** 4
**Confidence:** 3

**Summary:**

This paper proposes the Adapter-aware Structured Sparsification (ASSP), to prune the adapter-tuned SAM. This paper proposes the Projected-Gradient Residual (PGR) score to measure the significance of each module in the adapter-tuned SAM. PGR guides the pruning strategy. This paper evaluated ASSP's performance on the SAM-Med2D test set, showing that ASSP's performance is comparable to the original SAM-Med2D's performance when 76% of the parameters are pruned.

**Strengths:**

This paper proposed the PGR score to measure the significance of each adapter-tuned SAM module. PGR computes the residual of the model's gradients projected on the upstream and downstream gradient plane. The larger residual, the more important that module.

The experimental results show the effectiveness of the proposed ASSP. By comparing with the original SAM-Med2D, when 76% of the parameters are pruned, ASSP's performance is still comparable to SAM-Med2D's performance. In addition, ASSP shows superior performance to the SlimSAM approach.

**Weaknesses:**

Some parts of the papers are not clear or hard to understand.
- The PGR scores are defined for the gradients. However, in each training iteration, the units' gradients could be different, and so are the computed PGR scores for the units. How to compute the final PGR scores for units?
- In line 5 of Algorithm 1, what is the criterion to partition row indices into disjoint sets?
- In Sec. 3.2, many concepts are hard to understand. For example, "structured co-keep/co-prune constraint", "operator-dimension matching", "residual-add shape alignment", "Global dependencies", "global coupling", and so on.


It's better to show some qualitative segmentation results.

**Questions:**

In the experiments, does the proposed method ASSP start from the weights of the pretrained SAM-Med2D?

---

### Official Review · Reviewer_B32k · 2025-11-01

**Soundness:** 3
**Presentation:** 3
**Contribution:** 2
**Rating:** 2
**Confidence:** 4

**Summary:**

The authors introduce an adapter-aware structured pruning framework that incorporates a two-stage dual-stream compensation mechanism. In addition, they propose a Projected Gradient Residual criterion designed to align the gradient subspaces between upstream and downstream tasks.

**Strengths:**

1. The authors identify that the frozen backbone is more sensitive to pruning compared to the adapters.
2. They propose an adapter-aware structured pruning framework tailored to this observation.
3. They introduce a Projected Gradient Residual criterion to effectively fuse the upstream and downstream gradient subspaces.
4. They design a two-stage dual-stream compensation scheme to enhance performance after pruning.

**Weaknesses:**

1. The proposed method is restricted to adapter-tuned SAM models, while more widely adopted variants such as MedSAM are not supported. Since many SAM-based models do not employ adapters, this limitation raises concerns about the general applicability of the approach. The experiments are conducted solely on SAM-Med2D, further constraining the scope of validation.

2. The authors are encouraged to evaluate their method on additional adapter-based SAM variants, such as 3DSAM-Adapter, MA-SAM, or MaskSAM, to demonstrate broader applicability. However, many adapter-tuned SAM models (e.g., 3DSAM-Adapter, MA-SAM, MaskSAM) already contain relatively few parameters, which may limit the potential benefits of additional pruning.

3. Moreover, the experiments are limited to a single dataset, making it difficult to assess the method’s generalization across diverse data distributions. Finally, the absence of an ablation study for the proposed Projected Gradient Residual (PGR) criterion leaves its individual contribution to the overall performance unclear.

[1] Gong, Shizhan, et al. "3dsam-adapter: Holistic adaptation of sam from 2d to 3d for promptable medical image segmentation." arXiv e-prints (2023): arXiv-2306.
[2] Chen, Cheng, et al. "Ma-sam: Modality-agnostic sam adaptation for 3d medical image segmentation." Medical Image Analysis 98 (2024): 103310.
[3] Xie, Bin, et al. "MaskSAM: Towards Auto-prompt SAM with Mask Classification for Volumetric Medical Image Segmentation." arXiv preprint arXiv:2403.14103 (2024).

**Questions:**

1. Clarify whether the proposed method can be extended beyond adapter-tuned SAM models to more widely used variants such as MedSAM.
2. Demonstrate how the approach would apply to SAM-based models that do not incorporate adapters.

3. Evaluate the proposed method on additional adapter-based SAM variants (e.g., 3DSAM-Adapter, MA-SAM, MaskSAM) to validate its general applicability.

4. Discuss the potential benefits of pruning in adapter-tuned SAM models that already contain few parameters.

5. Extend the experimental evaluation to multiple datasets to assess generalization across diverse data distributions.

6. Conduct an ablation study to isolate and quantify the contribution of the Projected Gradient Residual (PGR) criterion to the overall performance.

---

### Official Review · Reviewer_xe5R · 2025-11-03

**Soundness:** 3
**Presentation:** 3
**Contribution:** 3
**Rating:** 4
**Confidence:** 2

**Summary:**

This paper proposes Adapter-aware Structured Sparsification (ASSP), a structured pruning framework for adapter-tuned Segment Anything (SAM) models, i.e., SAM-Med2D. In detail, ASSP introduces a Projected Gradient Residual (PGR) scoring criterion that merges upstream (pretraining) and downstream (fine-tuning) gradient subspaces to identify redundant structures. The method performs pruning and compensation in two sequential stages: first the adapters, then the backbone, with a dual-stream training scheme that alternates supervision between pretraining and downstream datasets. Experimental results suggest that ASSP can significantly reduce compute while maintain Dice score performance, and outperform SlimSAM baselines.

**Strengths:**

1. The dependency analysis is well-motivated and technically sound.

2. The PGR criterion is a novel and theoretically justified improvement over traditional magnitude or Taylor-based pruning.

**Weaknesses:**

1. The title of this paper seems to imply ASSP is a general method. However, this paper is limited to SAM-Med2D and medical imaging area. Thus, the effectiveness on other domains and architecture is unclear.


2. This paper only compares to SlimSAM. If possible, other efficient SAM methods, not limited to medial area, should be included.

**Questions:**

1. How does ASSP perform when applied to non-medical SAM variants (e.g., natural images or remote sensing)? Are the pruning dynamics consistent across domains?

2. Can the authors quantify the additional computational cost of the PGR computation and dual-stream fine-tuning relative to standard pruning baselines.

---

### Note · Authors · 2025-11-14

I have read and agree with the venue's withdrawal policy on behalf of myself and my co-authors.